# Ethnic Identity and Collective Self-Esteem Mediate the Effect of Anxiety and Depression on Quality of Life in a Migrant Population

**DOI:** 10.3390/ijerph19010174

**Published:** 2021-12-24

**Authors:** Alfonso Urzúa, Diego Henríquez, Alejandra Caqueo-Urízar, Rodrigo Landabur

**Affiliations:** 1Escuela de Psicología, Universidad Católica del Norte, Antofagasta 1240000, Chile; xdiegohenriquez@gmail.com; 2Instituto de Alta Investigación, Universidad de Tarapacá, Arica 1000000, Chile; acaqueo@academicos.uta.cl; 3Departamento de Psicología, Universidad de Atacama, Copiapó 1530000, Chile; rodrigo.landabur@uda.cl

**Keywords:** quality of life, migration, mental health, anxiety, depression, ethnic identity

## Abstract

Migration processes can often trigger negative interactions with the context, generating problems in both the physical and mental health of migrants, which have an impact on both their well-being and their quality of life. In this framework, the research aimed to assess the mediating effect of ethnic identity and collective self-esteem on the inverse relationship between mental health problems and quality of life. Data were collected from 908 first-generation Colombian migrants living in Chile, of whom 50.2% were women and with an average age of 35 years. They were assessed with The World Health Organization Quality of Life (WHOQoL)-Bref, Smith’s ethnic identity questionnaire, Basabe’s collective self-esteem, and Beck’s Anxiety and Depression questionnaires. The results provide evidence that both collective self-esteem and ethnic identity exert a partial mediation effect on the relationship between anxiety and depression on quality of life. The study provides evidence of the protective role that the maintenance and reinforcement of ethnic identity and collective self-esteem can have, with a view to strengthening the planning of interventions both in the field of prevention of mental health problems and in the improvement of quality of life based on evidence.

## 1. Introduction

An international migrant is a person who moves outside their usual place of residence, across an international border, temporarily or permanently, for a variety of reasons [1]. In 2020, the global migrant population was 272 million people or 3.5% of the world’s total population [2]. There is abundant evidence that migration processes involve changes both in the people who migrate and in the societies that host them and with which they interact. At the individual level, the health and well-being of migrants can be negatively affected by phenomena such as discrimination and prejudice, affecting and diminishing their quality of life [3].

Quality of life (QoL) is an indicator of well-being. At an operational level, it can be defined as the level of satisfaction and well-being in several domains of daily life [4] that is the product of a cognitive evaluation process based on subjective standards of comparison [5]. In a non-diseased migrant population, there is some evidence of a relationship between perceived quality of life and age, gender, income level, close relationships, the process and strategies of acculturation, social support, discrimination, and stress [3,6,7,8,9,10,11,12,13,14,15,16].

The relationship between the presence of mental health problems and QoL has been extensively studied in the field of severe mental disorders. There is little research on the relationship between mental health and QoL in the general population [17,18,19] and even less in the migrant population [9]. Despite this, it is likely that, in this population, QoL as an indicator of well-being is negatively affected by the presence of mental health problems, a relationship that has been found in other closely related indicators such as subjective well-being [20], psychological well-being, and life satisfaction [21,22].

This research focuses on south-south migration, i.e., people moving from one South American country to another, and in this particular case, Colombians to Chile. Our research group has provided evidence of the influence of negative factors such as discrimination on both their QoL and mental health, as well as reporting the protective effect that ethnic identity and collective self-esteem seem to have on the negative effects that discrimination has on well-being, QoL, or mental health [23,24,25,26,27]. In this context, the aim of the present research is to evaluate the mediating effect that both ethnic identity and collective self-esteem may have on the inverse relationship between the presence of symptoms associated with mental health problems and QoL. We believe that there could be a mediating effect of ethnic identity and collective self-esteem since it has been found that these variables have been able to absorb the negative effect that various variables may have on indicators of well-being in other studies in a similar population [23,26,28], possibly exercising a protective role through the strengthening of individual self-esteem, which could be extrapolated to the case of quality of life.

The following hypotheses have been put forward as hypotheses: H1: Anxiety has a negative effect on collective self-esteem, QoL, and ethnic identity; H2: Depression has a negative effect on collective self-esteem, QoL, and ethnic identity; H3: Anxiety has a positive relationship with depression; H4: Collective self-esteem mediates the effect of anxiety on QoL; H5: Collective self-esteem mediates the effect of depression on QoL; H6: Ethnic identity mediates the effect of anxiety on QoL; and H7: Ethnic identity mediates the effect of depression on QoL.

## 2. Materials and Methods

### 2.1. Participants

A total of 908 first-generation Colombian immigrants residing in Chile were surveyed. They were contacted in various places such as the Department of Foreigners and Migration, Jesuit Migrant Service, health centers, workplaces, and neighborhoods with a high number of migrant inhabitants. The type of sampling is non-representative; it was combined, including snowball sampling with targeted sampling in places with a high influx of migrants. Inclusion criteria were being a Colombian resident in Chile for more than 6 months and over 18 years of age. Data were collected from three main cities with migrants in Chile: Antofagasta 461 (50.8%), Arica 222 (24.4%), and Santiago 225 (24.8%). Of the total number of respondents, 50.2% were women. The age range of participants was between 18 and 89 years, with a mean age of 36 years (SD = 10.23). The sociodemographic characterization of the participants can be seen in Appendix A (Table A1).

### 2.2. Measures

#### 2.2.1. Quality of Life

We used a Spanish version [29,30] of the WHOQoL-BREF [30], which has been used in both Chilean and South American populations, reporting adequate psychometric properties for its use [31,32,33]. This instrument consists of 26 questions, grouped into 4 dimensions: physical (7 items), psychological (5 items), social (3 items), and environmental (8 items). The use of this instrument was authorized by the Mental Health Division of the World Health Organization.

#### 2.2.2. Ethnic Identity

To assess this identity variable, the Ethnic Identity dimension of the adaptation of the “Ethnic Multigroup Identity Scale” (EIEM) developed by Phinney [34], in its version for Spanish-speaking populations in Latin American countries [35], was used. This dimension assesses the importance, affective meaning, and value connotation of belonging to a group (e.g., “I am aware of my ‘ethnic roots’ and what they mean to me”, “I am very proud of my ethnic group”). The response option is a four-point Likert-type, where high scores reflect a strong, positive orientation towards the reference ethnic group. This scale has been previously used in studies of migrant populations in Chile, reporting adequate reliability [24,36]. The questions are based on the ethnic identity proposal of Tajfel [37], who conceives them as part of an individual’s self-concept that derives from the knowledge of their belonging to a social group (or social groups) together with the evaluative and emotional meaning associated with said belonging. In this context, the questions were directed directly to his identity as a Colombian (e.g., I am happy to be Colombian; I feel very close to my country and its culture).

#### 2.2.3. Collective Self-Esteem

To assess this variable, 4 items from the scale proposed by Luthanen and Crocker [38] were used. These ask about how the person felt in relation to his or her national group of origin. This scale has been previously used in the South American migrant population reporting adequate psychometric properties [27].

#### 2.2.4. Depression

We used the BDI IA Inventory, which assesses the self-report of depressive symptomatology. This questionnaire was developed by Beck [39]; it has a revised version in 1978, translated into Spanish [40]. This instrument has adequate psychometric properties [41]. The internal consistency for the present study was a Cronbach’s alpha of 0.93.

#### 2.2.5. Anxiety

Anxiety was assessed using the Beck Anxiety Inventory BAI, which asks about common symptoms associated with anxiety disorders. The Spanish version [42,43] of the Beck Questionnaire [44] was used in this study. In the present study, the internal consistency measured by Cronbach’s alpha was 0.94 for the Colombian population and 0.95 for the Chilean sample.

### 2.3. Procedures

This research is part of a larger project on discrimination and health, which was reviewed and approved by an accredited Scientific Ethics Committee under resolution 011/2018. All participants signed an informed consent form prior to participation. Once the data were collected, they were entered into a database constructed for this purpose in SPSS.25 (IBM Corp., Armonk, NY, USA).

### 2.4. Data Analysis

First, measurement models for each of the constructs were estimated through confirmatory factor analysis (M1, M2, M3, M4, and M5). Subsequently, the first hypotheses (H1, H2, and H3) were tested using three structural models. The first model (M6) estimated the effect of anxiety (ANX) on collective self-esteem (CS), QoL, and ethnic identity (EI). The second model (M7) estimated the effect of depression (DEP) on collective self-esteem, QoL, and ethnic identity. The third model (M8) estimated the covariation between anxiety and depression. Once these hypotheses were tested, we proceeded to test the mediation hypotheses (H4, H5, H6, and H7) through a structural multiple mediation model (M9), where we estimated the mediating effect of collective self-esteem and ethnic identity on the effect of anxiety and depression on QoL. The indirect effects of the mediation model were estimated following the recommendations of Stride et al. [45]. In all analyses, the effects of age, sex, and years of stay in Chile were controlled for. The goodness-of-fit of all structural models was estimated using chi-square (χ2) values, root mean square error of approximation (RMSEA), comparative fit index (CFI), and Tuker–Lewis index (TLI). The maximum likelihood robust (MLR) estimation method was used for all analyses, which is robust to non-compliance with the multivariate normality assumption [46]. The statistical packages used were SPSS v. 25 [47], Jamovi v. 0.9 (Jamovi Project, Sydney, Australia), and MPlus v. 8.2 (Muthen & Muthen, Los Angeles, CA, USA). The descriptive statistics of each of the variables incorporated in the various models can be reviewed in Table A2.

## 3. Results

### 3.1. Measurement Models

The measurement models for the measures of QoL (M1), collective self-esteem (M2), ethnic identity (M3), anxiety (M4), and depression (M5) presented acceptable goodness of fit, with indices close to the standards recommended by the literature (RMSEA < 0.08; CFI > 0.95; TLI > 0.95) [48]. Table 1 shows the goodness-of-fit indices of the measurement models.

Cronbach’s alpha reliability coefficient was also estimated for each of the dimensions of the measurement models. These are for collective self-esteem 0.73, ethnic identity 0.85, anxiety 0.92, and depression 0.88. For the QoL dimensions: 0.65 (social), 0.72 (physical), 0.80 (psychological), and 0.66 (environmental).

### 3.2. Structural Equation Models

With the measurement models and reliability coefficients estimated, we then proceeded to evaluate the first hypothesized structural models: M6 (Figure 1), M7 (Figure 2), and M8 (Figure 3). Table 2 shows the goodness-of-fit indices of these models.

In M6 (Figure 1), it can be observed that anxiety exerts a small negative effect (b > 0.10) [49] on collective self-esteem (b = −0.125), and a moderate negative effect (b > 0.30) on QoL (b = −0.306) and ethnic identity (b = −0.333).

M7 (Figure 2) shows that depression has a small negative effect on collective self-esteem (b = −0.171) and ethnic identity (b = −0.191) and a moderate negative effect on QoL (b = −0.287).

Finally, in M8, we found that anxiety and depression maintained a moderate positive relationship (b = 0.373).

### 3.3. Multiple Mediation Model

Having observed the relationships posited in models M6, M7, and M8, a structural multiple mediation model (M9) was tested to test hypotheses H4, H5, H6, and H7. In the structural multiple mediation model (Figure 3), it can be observed that collective self-rsteem exerts a partial mediation effect [50] on the relationship between anxiety and QoL. Similarly, it can be observed that ethnic identity exerts a partial mediation effect on the relationship between depression and QoL. In addition, it can be observed that the joint presence of collective self-esteem and ethnic identity presents significant total indirect effects on the relationship between anxiety, depression, and quality of life.

The proposed structural multiple mediation model (M9) presented goodness-of-fit levels close to the criteria accepted by the literature (RMSEA = 0.039; CFI = 0.922; TLI = 0.916) (Table 2).

Table 3 shows the standardized indirect and total indirect effects of the mediation model presented in Figure 3. Here it can be seen that, regarding the indirect effects of CS and EI, these present differentiated effects. On the one hand, the EI presents a significant mediation effect only between the ANX and the QoL, while the CS does so only in the relationship between DEP and QoL. Additionally, when both variables are presented simultaneously (EI and CS), they also manage to mediate the negative effect of anxiety and depression on QoL (indirect total effect).

## 4. Discussion

The aim of the present research was to assess the mediating effect that both ethnic identity and collective self-esteem might have on the inverse relationship between the presence of symptomatology associated with mental health problems and QoL. As we proposed, both anxiety and depression had a negative effect on collective self-esteem, quality of life, and ethnic identity. Anxiety and depression were also positively associated with each other. As for the mediation effects, contrary to what we hypothesized, collective self-esteem did not mediate the effect of anxiety on QoL, but it did mediate the effect of depression on QoL. Ethnic identity mediated the effect of anxiety on QoL but did not mediate the effect of depression on QoL.

The main contribution of this study is to provide evidence on the protective role of collective self-esteem and ethnic identity on the negative effects of mental health problems on QoL, depending on the type of symptomatology. For anxious symptoms, their negative effects on QoL are reduced by ethnic identity. This result is consistent with the protective effect of ethnic identity on the relationship between anxiety and emotionally influenced food consumption [24] and on the relationship between discrimination and well-being [23], whereas for depressive symptomatology, its negative effect on QoL can be diminished through collective self-esteem. This result should be understood in relation to previous studies indicating an increase or decrease in collective self-esteem because of the type of perceived discrimination, racial or ethnic [25]. Thus, the present study enriches the literature by analyzing the specific mechanisms by which discrimination would impact QoL and how to reduce these negative effects. This is because some of the anxious and depressive symptomatology may be due to perceived discrimination [25].

The QoL of the immigrant population can be approached from the perspective of both the immigrant and the local population. In the first case, future studies can enhance the protective factors of collective self-esteem and ethnic identity, considering that our results show an important direct influence of anxious and depressive symptomatology on QoL. Specifically, these factors can be enhanced through the collective activities of the immigrant population, as these activities are associated with greater collective self-esteem and identity connection with the group and its members [51].

The QoL of the immigrant community can also be approached from the local population, promoting in the latter the inclusion of their national and immigrant identity within a higher category. This is proposed by the Common Endogroup Identity Model, where the process of recategorization into a more inclusive identity, e.g., a Latin American or global-universal identity, decreases intergroup bias and thus discrimination towards the immigrant population [52,53]. This would imply the development of joint activities between the local and immigrant community to create spaces for this more inclusive identity to take place and be promoted.

Regarding the limitations of the present study, this study covered the Colombian immigrant population in three cities, so these results cannot be directly extrapolated to other migrant populations in other sectors. Future studies could replicate these results in other regions where this immigrant community has increased, and in immigrant populations with a higher incidence in recent years, such as those from Haiti and Venezuela [54], or analyze the effect that living in immigrant neighborhoods (which would favor the maintenance of identity and collective self-esteem) could have on the relationships studied versus living in neighborhoods with a low density of migrants. Another limitation is the cross-sectional measurement of variables, which prevents us from referring to causal relationships. This can be addressed with longitudinal measures, which would include discrimination and its impact on QoL, together with the mediating and moderating variables of this relationship. Protective mediating variables would include collective self-esteem and ethnic identity, the assessment of which can consider the empowering impact of engaging in collective activities detailed above.

## 5. Conclusions

The results provide evidence that both collective self-esteem and ethnic identity exert a partial mediation effect on the relationship between anxiety and depression on quality of life. These findings are relevant inputs for public policies to prevent mental health problems and mitigate their negative effects on the QoL of the immigrant population. Such policies should consider, among other factors, the relationship of the migrant community with their nation of origin and the local community, aiming at inclusion in the host society where different identities can live in a healthy coexistence that respects and welcomes these differences.

## Figures and Tables

**Figure 1 ijerph-19-00174-f001:**
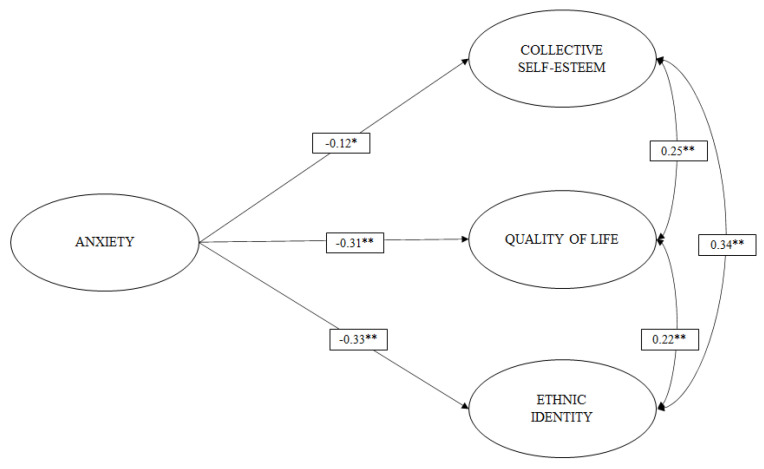
M6 Structural model where anxiety has an effect on collective self-esteem, ethnic identity, and QoL. The analysis controlled for the effects of years of stay, sex, and age. * *p* < 0.05; ** *p* < 0.001.

**Figure 2 ijerph-19-00174-f002:**
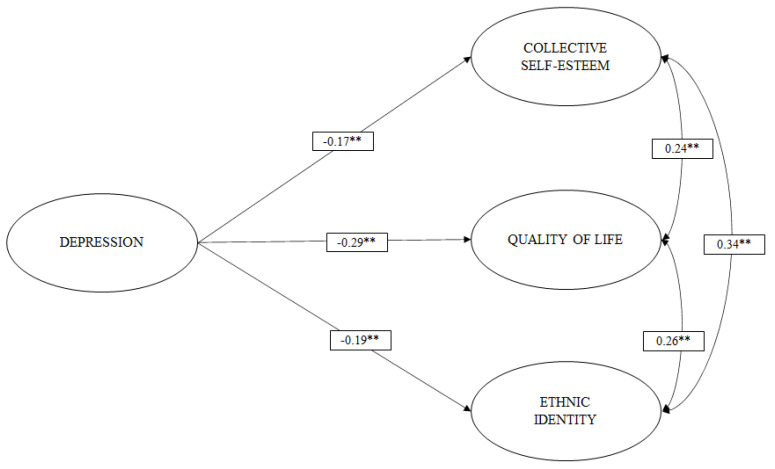
M7 structural model where depression exerts an effect on collective self-esteem, ethnic identity, and QoL. The analysis controlled for the effects of years of stay, sex, and age. * *p* < 0.05; ** *p* < 0.001.

**Figure 3 ijerph-19-00174-f003:**
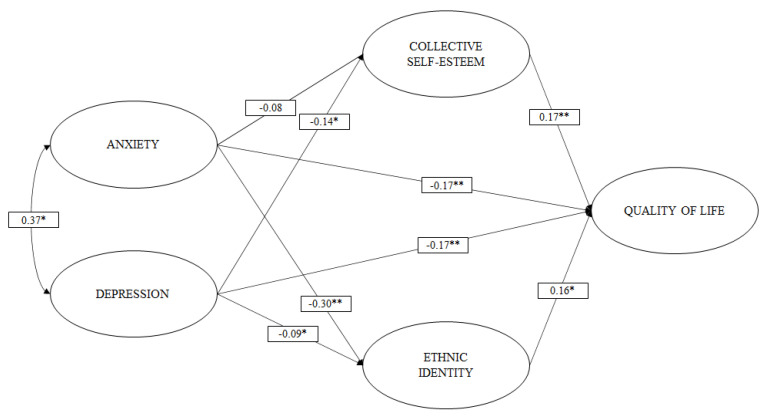
M9 structural model of mediation. * *p* < 0.05; ** *p* < 0.001.

**Table 1 ijerph-19-00174-t001:** Indicators of overall fit of the measurement models and estimated structural models.

Models	Parameters	χ^2^	DF	*p*	CFI	TLI	RMSEA	RMSEA CI 90%	
								Low	Sup
M1	40	198.985	50	0.00	0.941	0.922	0.057	0.049	0.066
M2	12	15.776	2	0.00	0.968	0.905	0.087	0.051	0.129
M3	15	33.084	5	0.00	0.968	0.936	0.079	0.055	0.105
M4	24	59.733	20	0.00	0.981	0.973	0.047	0.034	0.061
M5	24	44.536	20	0.00	0.977	0.968	0.037	0.022	0.052

M1—quality of life (QoL) measurement model; M2—collective self-esteem (CS) measurement model; M3—ethnic identity (EI) measurement model; M4—anxiety (ANX) measurement model; M5—depression (DEP) measurement model.

**Table 2 ijerph-19-00174-t002:** Indicators of overall fit of the estimated structural models.

Models	Parameters	χ2	DF	p	CFI	TLI	RMSEA	RMSEA IC 90%	
								Low	Sup
M6	10697	1061942.03100	367445	0.00	0.935931	0.928924	0.0401	0.038037	0.045043
M7	97106	9411058.310887	367445	0.00	0.922918	0.914909	0.042040	0.038036	0.045043
M8	4961	241335.423963	103148	0.00	0.965960	0.960954	0.039037	0.032	0.045043
M9	124127	14841646.885538	616724	0.00	0.922918	0.916912	0.039038	0.037036	0.042040

M6—effect of ANX on CS, QoL, and EI; M7—effect of DEP on CS, QoL, and EI; M8—relationship between ANX and DEP; M9—model where ANX and DEP are independent variables (IV), QoL is a dependent variable (DV), and CS together with EI are mediators between IV and DV.

**Table 3 ijerph-19-00174-t003:** Standardized indirect and total indirect effects of the mediation model.

	M9		
Effects	Indirect	Total Indirect	Total
ANX→EI/CS→QoL		−0.07 **	−0.24 **
ANX→EI→QoL	−0.05 *		
ANX→CS→QoL	−0.02		
DEP→EI/CS→QoL		−0.04 *	−0.22 **
DEP→EI→QoL	−0.02		
DEP→CS→QoL	−0.02 *		

QoL—quality of life; ANX—anxiety; DEP—depression; EI—ethnic identity; CS—collective self-esteem. * *p* < 0.05; ** *p* < 0.001.

## Data Availability

The data presented in this study are available on request from the corresponding author. The data are not publicly available because the project has state funding and will only be released once the project is finished.

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
