# Peer review of "Ethnic Identity and Collective Self-Esteem Mediate the Effect of Anxiety and Depression on Quality of Life in a Migrant Population"

_ijerph, 2021, doi:10.3390/ijerph19010174_

Round 1

Reviewer 1 Report

The paper assesses the mediating effect of ethnic identity and collective self-esteem on the inverse relationship between anxiety/depression and quality of life. Authors use a cross-sectional survey dataset collected by interviewing 908 first-generation Columbian migrants in Chile. The main finding is that ethnic identity mediates the effect of anxiety, while collective self-esteem mitigates the effect of depression.

The paper addresses an important question. It is well-organized and well-written. I also appreciate that the authors are open about limitations of their dataset and do not claim causality of their findings. The paper is publishable if the authors can provide more background behind their hypotheses, explain methodology, and add interpretations to their results.

  • Why can ethnic identity and collective self-esteem mediate the effect of anxiety and depression on QoL? It would be very helpful if the authors can give examples for potential mechanisms they have in mind (Introduction, before formulating the hypotheses).
  • It seems that the authors collected a very interesting dataset. They provide some descriptive statistics in Section 2.1, but it would help to have a summary table (in the Appendix) with baseline characteristics of migrants (gender, age, time since arrival in Chile, education?, employment?) and their responses (mean, st. deviation, min, max, number of non-missing responses). I understand that QoL, depression, anxiety, ethnic identity and self-esteem are composite indices, but it would still help to get the feeling of averages, variance, and number of non-missing observations
  • It was not clear to me what was meant under ethnicity: being Columbian, being Latin-American? How was the question formulated?
  • I would appreciate more interpretation of the results. In particular, how can one interpret the values for Cronbach's alpha reliability coefficient? How can one link the coefficients in Figure 3 to those in Table 3? How can one interpret the coefficients in Table 3? Are they not only statistically, but also economically significant?
  • While the authors acknowledge that they do not capture causal effects, it would be interesting to have a small discussion on what exactly could bias the results. What could influence both ethnic identity/collect self-esteem and QoL: time since arrival, living within an immigrant community?

Author Response

Dear reviewer, in the attached file we explain how we have resolved each of your reviews.

Reviewer 2 Report

The paper is very well presented and organised. A minor suggestion is to comment further the Results section since it deserves more attention in the overall balance of the article.

Author Response

(The authors gave the same response as above.)
